# Risk Factors for Collisions and Near-Miss Incidents Caused by Drowsy Bus Drivers

**DOI:** 10.3390/ijerph17124370

**Published:** 2020-06-18

**Authors:** Genta Miyama, Masakatsu Fukumoto, Ritsuko Kamegaya, Masahito Hitosugi

**Affiliations:** 1Department of Legal Medicine, Shiga University of Medical Science, Shiga 520-2192, Japan; hitosugi@belle.shiga-med.ac.jp; 2i·OH Laboratory Co., Ltd., Tokyo 153-0063, Japan; ae86fm@hotmail.co.jp; 3Health Service Planning, Tokyo 160-0022, Japan; ritsuko.kamegaya@healthservice-p.net

**Keywords:** drowsy driving, risk factors, bus drivers, safety education, risk management

## Abstract

Serious accidents have been caused by drowsy bus drivers and have necessitated an examination of the risk factors involved. A questionnaire survey among employees of a bus company was conducted in Ibaraki Prefecture, Japan in September 2014. Respondents were asked to report details of their work and life over the preceding month. The 301 valid responses by bus drivers (295 men and 6 women) with a mean age of 51.6 years (range: 24–73 years) were used for analysis. Univariable logistic regression showed that factors affecting the incidence of collisions and near-miss incidents by drowsy drivers were continued driving when feeling sick, reporting a physical condition, number of sleep hours, time spent with family, working hours, and nutritional balance. According to a multiple regression analysis, continued driving when feeling sick (odds ratio: 3.421, 95% confidence interval: 1.618–7.231) was the only significant risk for the event. Managers should encourage drivers to voluntarily report poor health and should provide opportunities to stop driving if drivers experience physical discomfort or sleepiness. To improve road safety, educational measures are required for both drivers and managers to prevent driving under poor health conditions, although the decision to stop driving depends on drivers’ subjective judgment.

## 1. Introduction

As part of regulatory reform in public transportation, Japan’s Ministry of Land, Infrastructure, Transport, and Tourism (MLIT) deregulated the chartered-bus, route-bus, and highway-bus business in 2000, hoping to revitalize the industry. According to MLIT statistics, deregulation was immediately followed by a near doubling of the number of new entrants, including those from other sectors, such as taxi operators, into the chartered-bus business [1]. Additionally, online reservations increased with the development of information-delivering means on buses; the opportunities for brand recognition by potential users have resulted in the growth of the chartered-bus business [2]. Efforts to improve service and optimize efficiency have contributed to the revitalization of the industry [1].

However, deregulation has created excessive competition among operators, which has led to a worsening working environment for bus drivers. Major crashes, including a tour-bus crash with seven fatalities on an expressway in Gunma Prefecture in 2012 [3] and a ski-bus crash with 15 fatalities including two bus drivers in Nagano Prefecture in 2016 [4], have raised public awareness regarding the dangers of drowsiness in bus drivers. Subsequently, both the public and private sectors have undertaken various measures to improve drivers’ working conditions. For example, in 2016, MLIT mandated that bus operators record and store video footage of accidents and near-miss incidents using a drive recorder [5]; this footage is used for driver education and to determine the causes of these incidents during the ensuing investigations. In 2018, MLIT revised its safety regulations on commercial passenger-transportation businesses to add a provision that requires drivers to report lack of sleep in a roll call prior to driving [6]. According to a systematic review, drowsiness and fatigue significantly impair driving performance and expose drivers to increased risk [7]. In the United States, drowsy drivers account for an estimated 16.5% of fatal crashes and 13.1% of crashes leading to hospitalizations [8]. Thus, the prevention of driving while drowsy is an urgent public health issue. However, the risk factors leading to drowsiness-related driving accidents remain to be comprehensively identified.

Various potential technological solutions for reducing the load on bus drivers have been developed in recent years. One potential aid is the introduction of automated vehicles. In Japan, Tokyo Electric Power Company Holdings (TEPCO) introduced a self-driving bus in 2018 as a means of transporting field workers at the Fukushima No. 1 Nuclear Power Plant [9]. However, this bus runs only on preprogrammed routes. Regarding the introduction of self-driving vehicles on public roads, MITI plans to revise relevant laws for setting up priority lanes exclusively for self-driving vehicles on some sections of expressways [10]. On highways, offering point-to-point services would be a solution in which autonomous vehicles follow the routes instructed by the system’s control center; the control center assigns transportation requests to the appropriate self-driving vehicles [11]. The impact of advancements in mobility with the introduction of autonomous vehicles would be visible on highways connecting urban areas; however, this system would not be beneficial in rural areas [12] in which irregular roads and circumstances that may confound the sensing techniques of automated vehicles are prevalent. Furthermore, when a takeover is required, the drivers must drive the vehicle manually. 

There are still many problems to be solved in the introduction of self-driving vehicles. Azmat et al. [12] noted the threat of cybercrime. Communication between devices is achieved with the help of sensors, tags, and radio frequency identification; these endanger data authentication and integrity as this kind of wireless communication remains vulnerable to cyberattacks. An online survey suggested that another hurdle in introducing self-driving vehicles is skepticism about the technology among consumers. Despite the high number of vehicle collisions, the subjective feeling of safety in conventional vehicles is very high among the general public. This distorted perception complicates the introduction of autonomous vehicles [13]. Therefore, despite developments in automated vehicle technology and systems, human drivers must still be responsible for safe vehicle driving for the next several years.

Another potential solution is the adoption of drowsiness prediction systems using Internet-based technologies. Recently, machine learning was applied to predict drowsiness and to improve drowsiness prediction using facial recognition technology, eye-blink recognition technology, and CO_2_-sensing technology [14]. Further development of technologies for objectively measuring the level of drowsiness of the driver and for informing them to stop driving and to rest at appropriate times can be expected in the near future. This technology may also contribute to reducing vehicle collisions, and some bus companies have already introduced such systems. However, these solutions involve a substantial cost for bus companies. Furthermore, some recently developed systems cannot be used for all roads. Therefore, the current study focused on steps that bus drivers and their companies can apply immediately rather than exploring potential future technological solutions.

For taxi drivers, the influence of several health status factors on the incidence of vehicle collisions has been investigated, including fatigue, drowsiness, and disease onset. One previous study revealed that fatigue management strategies to identify individual factors, business-related characteristics, and work environment factors are important components of road safety [15]. A study of taxi drivers indicated that an insufficient number of days off work and difficulty reporting poor health conditions were influential risk factors in collisions and near-miss incidents [16]. However, no previous studies have elucidated drowsiness-related factors that influence incidents caused by bus drivers. Identifying these risk factors would provide a valuable first step for formulating policies to reduce collisions or near-miss incidents by bus drivers. In the present investigation, a questionnaire-based survey among employees of a bus company was conducted in Ibaraki Prefecture, Japan to identify the risk factors that lead to drowsy driving.

To address the objectives of this study, measures for preventing vehicle collisions caused by drowsiness in bus drivers were investigated. Several recommendations are made for both bus drivers and bus company managers.

## 2. Materials and Methods

### 2.1. Subjects

The present study was designed to reveal the risk factors involved in drowsy driving and to identify the problems currently faced by bus drivers and their companies. To answer these research questions, a questionnaire survey was conducted. The questionnaire targeted all 609 employees—regular drivers, occasional drivers, and administrative staff—of a bus company operating chartered/highway/route buses in Ibaraki Prefecture, which is adjacent to metropolitan Tokyo. The survey was conducted in September 2014 using a non-anonymous questionnaire with a closed-ended format. The reason for the non-anonymity of the questionnaire was that the results were used for medical intervention by occupational physicians to improve the health status of employees. Respondents were asked to report details of their work and life in the month preceding the survey and to return the completed survey to their physician. The self-administered questionnaire with an explanation of the research objective included a statement assuring the participants of the survey’s anonymity. The protocol of the investigation and research was approved by the ethics committee of Dokkyo Medical University (Univ. 24002). 

### 2.2. Questionnaire Items

With reference to a safety education manual published by MLIT for passenger transport operators, such as bus, truck, and taxi companies [17], it was assumed that risk factors of drowsy driving would include drivers’ worsening working conditions, reduced sleep quantity and quality, and physical or mental stress. Therefore, the questions were designed using that assumption. The questionnaire covered 200 items grouped under eight categories, as follows.

1. General characteristics: age, sex, body mass index, marital status, years of driving-work experience, full-time/part-time employment, and type of bus driving (route/express/chartered)

2. Lifestyle: exercise, smoking, alcohol consumption, nutritional balance, medical conditions, and noticeable symptoms related to a physical disorder

3. Driving conditions during the preceding month: physical conditions during driving

4. Sleep during the preceding month: sleeping time, disorders such as sleep apnea (breathing stops and restarts during sleep)

5. Incidents/accidents related to drowsy driving during the preceding month: collisions or near-miss incidents and the causes of major accidents, if any

6. Stress during the preceding month, from both life and work

7. Fatigue during the preceding month: mental and physical factors causing fatigue

8. Requests for improvement of working conditions: 15 questions concerning worsening working conditions and their effects on physical or mental stress, reduced sleep quantity and quality, or lack of time spent with family.

### 2.3. Analysis Method

In analyzing the questionnaire data, the focus was only on drivers’ responses. To identify the most relevant risk factors for drowsy driving, both unilateral and multilateral regression analyses were conducted. Drivers were classified into two groups: an event group, who had experienced collisions or near-miss incidents because of drowsy driving, and a nonevent group, who had experienced no such incidents. In the current study, a logistic regression model was used to analyze risk factors. This model uses data to predict the likelihood of occurrence of a particular event and is useful for identifying risk factors. To identify the risk factors of drowsy driving, univariable and multivariable logistic regressions were conducted using IBM SPSS Statistics ver. 23 (IBM Corp., Armonk, NY, USA). *p* < 0.05 was considered statistically significant.

## 3. Results

### 3.1. Subject Characteristics

Valid responses were obtained from 519 of the 609 employees (85.2%); the number of drivers in that cohort was 301. Most of the drivers (295 of 301) were male. Mean age was 51.6 years (range: 24–73 years). Mean body mass index of the drivers was 24.3 kg/m^2^, and 73.8% were married. Mean number of years of driving buses was 20.2 (range: 0.1–54 years), and 78.2% of the drivers had more than 20 years of experience (Figure 1).

Table 1 lists the drivers’ travel distance per month, stratified by the type of bus driving. Half the respondents (50.9%) were full-time drivers, and 41.5% were part-time drivers. 

### 3.2. Lifestyle

More than half of respondents (54.2%) reported that they exercised regularly. Approximately one-third (34%) smoked, with an average of 17.4 cigarettes per day. More than half of drivers (56.8%) reported consuming alcohol, at an average of 4.4 times per week. The average amount of alcohol consumed was 1.5 *go* (about 270 mL), and the interval between ending alcohol consumption and beginning driving work was 9.8 h. Some drivers (14.6%) reported skipping breakfast, while 42.5% ate three meals regularly each day.

Figure 2 shows the prevalence of diseases and medical conditions reported by drivers (multiple answers allowed). In all, 167 drivers (55.5%) reported at least one current condition, 108 of them (64.7%) attended a hospital at the time of the survey, and almost all of them told their physicians about their driving duties. Approximately 41% of drivers reported taking medication, and three reported that the medication affected their driving. The physical symptoms most reported by drivers were back pain (51%) and stiff shoulders (53%). 

### 3.3. Driving Conditions during Preceding Month

Of 278 drivers, 37 (13.3%) had experienced poor physical conditions one or more times during driving. About 37% of drivers reported that they were reluctant to alert the company when feeling sick. When these drivers were asked what they did when experiencing poor physical conditions, some responded that they stopped driving (27.4%), continued driving (21.6%), or continued driving after using the bathroom (10.8%).

### 3.4. Sleep during Preceding Month

The sleep guidelines for health promotion by MLIT note that 7 h of sleep result in the lowest risk of lifestyle-related illness or death [17]. One study suggested that driving after sleeping 6 h or less and multiple driving sessions were significantly associated with both rear-end collisions and single-car accidents [18]. The drivers reported sleeping an average of 6.4 h, but 26.0% of the drivers reported sleeping less than 6 h and 23.8% of all drivers reported experiencing symptoms of sleep apnea. Additionally, 25.3% of drivers reported experiencing sleepiness sometimes or frequently. 

### 3.5. Incidents or Accidents Related to Drowsy Driving during Preceding Month

Near-miss incidents in the month preceding the survey were reported by 134 respondents (44.5%), and 35 drivers (11.7%) experienced collisions once or twice. The frequency of near-miss incidents was reported as “sometimes” by 34 drivers (11.4%) and “frequently” by two drivers (0.7%). When asked about the main causes of major accidents, 192 drivers cited heavy traffic (64.1%), poor working/break conditions (63.0%), lack of safety education (50.0%), irresponsible behavior by drivers (32.8%), and poor driving skills of drivers (31.8%).

### 3.6. Stress during Preceding Month from Both Life and Work

Two-thirds (68.5%) of the drivers reported that their lives were “always busy” or “often busy”, and 63.4% reported feeling a lack of time. Approximately 60% felt that they were busy because they had too much work, and 67.4% stated that consulting their supervisors regarding their workload would be difficult.

### 3.7. Fatigue during Preceding Month: Mental and Physical Factors Causing Fatigue

Nearly half of the respondents cited mental factors (48.5%) such as “worries”, “tension”, “lack of concentration”, and “lack of satisfaction” or physical factors (43.2%) such as “irregular working conditions”, “great burden associated with extra work”, and “late-night shifts” as the causes of their fatigue. One-third of the drivers (33.9%) complained that the amenities and facilities for rest breaks were not satisfactory. 

### 3.8. Requests for Improved Working Conditions

The respondents were given a list of 15 items related to working conditions and were permitted to select as many as they felt applied to their situation. The factors cited most often as adversely affecting the drivers’ working conditions were as follows: low salary (82.6%), difficulty in taking days off (71.7%), irregular working hours (71.6%), irregular meal times (70.2%), long working hours (69.4%), not enough social time with friends (63.1%), not enough time spent with family (56.4%), fatigue from work (53.1%), poor rest facilities (51.4%), inadequate bathroom (toilet) time (48%), not enough time for sleep (47.8%), poor ventilation (29.9%), and inadequate breaks (28.5%).

### 3.9. Analysis of Risk Factors

Univariable logistic regression was used to analyze the risk factors of drowsy driving between the event group and the nonevent group. Table 2 shows that continued driving when feeling sick, reporting a physical condition, inadequate sleep, inadequate time spent with family, long working hours, and nutritional balance were factors that affected the incidence of collisions and near misses.

Next, multiple logistic regression was conducted using a collision or near miss as a dependent variable. Independent variables were as follows: age, alcohol consumption, nutritional balance, poor physical conditions during driving, continued driving when feeling sick, reporting a physical condition, feelings of stress, irregular working hours, number of sleep hours, and time spent with family. These factors were selected if their differential values between the event (collision or near miss) and nonevent groups in univariable analysis were *p* < 0.1. Continued driving when feeling sick was found to be a significant risk factor in collisions or near misses (odds ratio (OR): 3.421, 95% confidence interval: 1.618–7.231; Table 3).

In the regression model, the coefficient of determination summarizes the proportion of variance in the dependent variable associated with the independent variables. In the present study, the pseudo-determination coefficient (Nagelkerke R-squared value) was 0.142. Additionally, the Wald test results revealed a significant probability of 0.0009, indicating that the model was statistically appropriate.

## 4. Discussion

To combat the increased number of accidents caused by driver health, MLIT issued a revised health-care manual for commercial drivers in 2016, citing in particular the need for ensuring adequate sleep time of at least 6 h [19]. In a previous study, it was found that insufficient sleep was a significant factor in incidents or accidents related to driving while drowsy [16]. Other relevant factors identified were overwork, chronic lack of rest, and drowsiness caused by consuming alcohol, taking medications, or sleep apnea [20].

Previous studies have reported that bus drivers experience drowsy driving, sleep problems, and low sleep quality [21,22,23]. In one study, approximately one in four bus drivers in Peru reported experiencing drowsy driving, and this factor increased the risk of collisions and near-miss incidents [24]. In another study, to determine the influence of sleep quality on drowsy driving, sleep quality and associated factors were investigated among Thai intercity bus drivers [25]. The results revealed that working night shifts (odds ratio of 20.6), rotating day and night shifts (OR of 17.0), alcohol consumption (OR of 2.7), being married (OR of 3.1), and not exercising (OR of 2.3) were related to poor sleep quality [25]. The authors of the study suggested that both the bus company and the individual should take steps to maintain healthy lifestyles and to improve working conditions [23]. On the basis of these previous studies, the present study examined the factors involved in drowsy driving that can cause collisions or near-miss events among bus drivers.

A previous study in commercial drivers in Japan suggested that bus drivers had a lower threshold for reporting sudden illness than taxi or truck drivers because of their belief in the importance of passenger safety [26]; however, it also found a tendency in the commercial drivers to continue driving under poor physical conditions. The current results revealed that continuing driving when feeling sick was a major risk factor for collisions and near-miss incidents. Therefore, increased education for both drivers and company managers is required. In a previous study, an analysis using a driving simulator suggested that an education program to increase knowledge of the impact of sleepiness on driving could reduce the risk of drowsy driving and associated road trauma in young drivers [27]. Bus companies should advise their drivers to report health conditions to supervisors and to refrain from driving when feeling unwell: the cooperation of the companies is essential. Managers should encourage drivers to voluntarily report poor health and should provide opportunities to stop driving when drivers feel physical discomfort or sleepiness. Overall, a working environment that encourages drivers to voluntarily and easily report health problems should be fostered.

Currently, annual health checkups are mandated for commercial drivers such as bus drivers, but drivers may or may not disclose personal information to employers. To prevent collisions, an environment that does not penalize drivers for reporting their physical and mental conditions should be fostered and more comfortable, well-equipped rest facilities should be provided. 

Mental stress resulting from dissatisfaction with inadequate family time was also a significant factor in the incidence of collisions and near misses. Increased competition in the chartered-bus industry has forced drivers to extend their working hours. A report by Japan’s Ministry of Internal Affairs and Communications noted that the average annual total working hours per bus driver was 1.3 times that of all industrial workers [28]. A recent study by our group in a cohort of taxi drivers found that insufficient vacation time was cited as the most influential risk factor for collision-related events, which indicated accumulated fatigue [16]. This reinforces the importance of cooperation by company managers in improving drivers’ work/life satisfaction and balance.

## 5. Conclusions

To identify risk factors for collisions and near-miss incidents caused by drowsiness among bus drivers, a questionnaire survey of employees of a bus company in Japan was conducted. Using univariable logistic regression and multiple regression analyses, it was shown that continuing to drive when feeling sick was a major risk factor for such incidents. 

The current study involved several limitations that should be considered. First, “drowsiness” and “near misses” were determined subjectively; therefore, the near misses based on drowsy driving may be underestimated. In a future study, near misses or drowsy driving should be determined objectively using driving recorders and onboard cameras to photograph drivers and to confirm the present results. Second, recent levels of fatigue and psychological conditions (within 1 month of the survey) were investigated. These factors should also be evaluated immediately after accidents or near-miss incidents. Third, the employees of only one company in one prefecture were surveyed; a cross-sectional survey of multiple companies and their regional characteristics will be necessary in the future. Fourth, drivers were surveyed in a non-anonymous manner, which may have biased the respondents toward underreporting near misses, physical conditions, and personal habits and problems. Fifth, this survey was conducted in September, immediately following a busy travel season, and many drivers complained that their physical and psychological loads were too high. Conducting a similar survey during the off-season may yield different findings.

Unfortunately, drivers were reluctant to report health conditions to their supervisors. Therefore, it is important for drivers and managers to have a better understanding of the risk factors that lead to sleep deprivation and subsequent accidents. Improving the work environment to maintain the mental and physical health of drivers could increase road safety.

## Figures and Tables

**Figure 1 ijerph-17-04370-f001:**
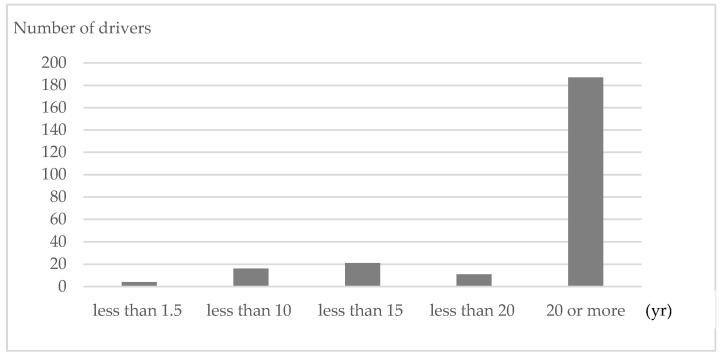
Driving experience of drivers in the survey.

**Figure 2 ijerph-17-04370-f002:**
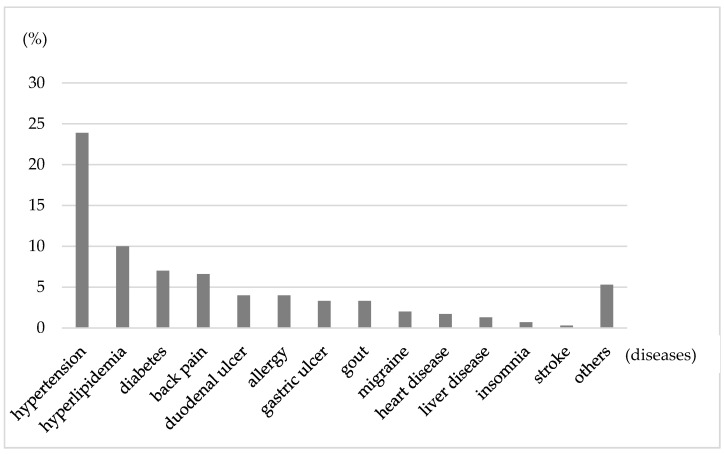
Prevalence of diagnosed diseases and medical conditions reported by bus drivers.

**Table 1 ijerph-17-04370-t001:** Average travel distance (km/month) by the type of bus driving.

Type of Bus Driving	Number of Drivers	Mean Mileage (SD)	
		Busy Season	Off-Season
	N = 301		
Route	186	2626 (2358)	2394 (1082)
Highway	18	5735 (1756)	5433 (890)
Chartered	14	5039 (2696)	3325 (1109)
Route and Highway	24	5963 (2093)	5862 (1877)
Route and chartered	10	5625 (2615)	2643 (1749)
All	14	7555 (3253)	5409 (2773)
Unknown	35	2181 (1520)	2083 (1157)

**Table 2 ijerph-17-04370-t002:** Univariable logistic regression analysis of risk factors for collisions and near misses.

	Event Group	Nonevent Group	Odds Ratio	95%CI	*p*-Value
	N = 170	N = 131			
Gender					
Male	167	128	Reference		
Female	3	3	0.766	0.151–3.896	0.748
Age (elderly person)					
Less than 65 yr	151	107	Reference		
65 yr or older	19	24	0.561	0.292–1.079	0.083
Marital status					
Married	124	98	Reference		
Unmarried	46	33	1.102	0.653–1.858	0.716
BMI					
Normal	103	74	Reference		
Low	5	3	1.197	0.275–5.211	0.622
High	62	54	0.825	0.513–1.325	0.425
Exercise habits					
No	81	57	Reference		
Yes	89	74	0.846	0.534–1.342	0.477
Smoking habits					
No	110	87	Reference		
Yes	60	44	1.079	0.666–1.748	0.758
Drinking habits					
No	66	64	Reference		
Yes	104	67	1.505	0.947–2.393	0.084
Eating breakfast					
No	25	21	Reference		
Yes	145	110	1.107	0.587–2.088	0.752
Eating habits					
Irregular	106	66	Reference		
Regular	64	65	0.613	0.385–0.976	0.039
Nutritional balance					
Poor	100	58	Reference		
Good	70	73	0.658	0.415–1.043	0.075
Having current or past illnesses					
No	91	55	Reference		
Yes	79	76	0.834	0.525–1.323	0.439
Feeling a poor physical condition while driving					
No	142	119	Reference		
Yes	28	12	1.955	0.949–4.029	0.069
Continued driving when feeling sick					
No	106	113	Reference		
Yes	64	18	3.790	2.102–6.835	0.000
Reporting a physical condition					
Easy	100	98	Reference		
Difficult	70	33	2.079	1.259–3.433	0.004
Feeling stress					
No	119	107	Reference		
Yes	51	24	1.911	1.098–3.325	0.022
Feeling free to talk to superiors					
No	117	79	Reference		
Yes	53	52	0.688	0.426–1.112	0.127
Feeling free to talk to colleagues					
No	118	90	Reference		
Yes	52	41	0.967	0.589–1.588	0.895
Irregular working hours					
Few times	92	87	Reference		
Many times	78	44	1.676	1.043–2.694	0.033
Number of sleep hours					
Enough	76	85	Reference		
Not enough	94	46	2.232	1.393–3.577	0.001
Time spent with family					
Enough	63	74	Reference		
Not enough	107	57	2.205	1.381–3.520	0.001

BMI, body mass index; CI, confidence interval.

**Table 3 ijerph-17-04370-t003:** Multivariable logistic regression analysis of risk factors for collisions and near-miss incidents related to drowsy driving (N = 301).

	Odds Ratio	95%CI	*p*-Value
Age (elderly person)			
Less than 65 yr	Reference		
65 yr or older	0.920	0.431–1.963	0.829
Drinking habits			
No	Reference		
Yes	1.575	0.963–2.576	0.070
Eating habits			
Irregular	Reference		
Regular	0.981	0.513–1.874	0.953
Nutritional balance			
Poor	Reference		
Good	0.879	0.473–1.632	0.681
Feeling a poor physical condition during driving			
No	Reference		
Yes	0.730	0.290–1.843	0.505
Continued driving when feeling sick			
No	Reference		
Yes	3.421	1.618–7.231	0.001
Reporting a physical condition			
Easy	Reference		
Difficult	1.002	0.556–1.805	0.995
Feeling stress			
No	Reference		
Yes	1.175	0.619–2.232	0.621
Irregular working hours			
Few times	Reference		
Many times	1.017	0.579–1.786	0.953
Number of sleep hours			
Enough	Reference		
Not enough	1.299	0.690–2.446	0.416
Time spent with family			
Enough	Reference		
Not enough	1.465	0.775–2.769	0.238

CI, confidence interval.

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
