# Peer review of "Risk Factors for Collisions and Near-Miss Incidents Caused by Drowsy Bus Drivers"

_ijerph, 2020, doi:10.3390/ijerph17124370_

Round 1
Reviewer 1 Report
The paper seems to have addressed most of the comments, yet I would still suggest good proofreading and sentence structure can be improved.
all the best.
Reviewer 2 Report
The authors have revised the manuscript. It is recommended for publication.
Author Response
Please see the attachment.

This manuscript is a resubmission of an earlier submission. The following is a list of the peer review reports and author responses from that submission.
Round 1
Reviewer 1 Report
Review of the manuscript # ijerph-801237
Title: Risk Factors for Collisions and Near-Miss Incidents Caused by Drowsy Bus Drivers
Comments / Feedback
The manuscript deals with an exciting. The authors have done a good job in empirically analyzing the risk factors. However, the depth and breadth of the topic catering drowsy behavior and impact of innovative technologies is missing. Overall, the paper demonstrates merit, as it addresses practical issues of day to day driving. Nonetheless, I would strongly suggest to that following comments shall be taken into account before publication in the journal.
1. In abstract you should consider adding limitations and implications for the policy makers, you have briefly suggested that but it needs to be more concrete.
2. The introduction is suitable for the most part; however, I would suggest including Research questions or argument based hypothesis towards the end of the introduction. You also need to address the need to study this topic and how would it help the scientific community or the industry.
3. The literature review is completely missing and I would suggest it is an important element in the paper of this sort to make reader understand the depth and breadth of the topic under discussion. A brief explanation driver behavior, socio-economic impact of accidents and a comparative analysis of advanced technologies with traditional technologies would be a good argument suggesting how things might change in future using autonomous vehicles and internet based solutions. It would be a good dea to forsee the introduction of Autonomus and Connected vehicles and asses the situation using these vehicles in futuristic settings. I would strongly recommend to see the following papers for theory building:
a. Azmat, M., Kummer, S. Potential applications of unmanned ground and aerial vehicles to mitigate challenges of transport and logistics-related critical success factors in the humanitarian supply chain. AJSSR 5, 3 (2020). https://doi.org/10.1186/s41180-020-0033-7.
b. Jang, S.-W.; Ahn, B. Implementation of Detection System for Drowsy Driving Prevention Using Image Recognition and IoT. Sustainability 2020, 12, 3037.
4. In its current form the papers lacks scientific soundness and is very blant without any theory building, reasoning and dispositions, I would strongly suggest to add a multi channel outlay to the research by adding variety of dimension to the literature by citing the above mentioned papers and similar papers in this category.
5. Paper’s methodology seems to be very basic and unclear at the same time, there is a need to address clearly:
a. what methodology has been selected and on what grounds.
b. Why certain testing was done and not simple regression?
c. Was data normally distributed or not?
d. Why data was not collected anonymously?
e. Data collected for this research is from 2014 – since then a lot has changed, how would you justify the credibility of the data.
f. I would strongly suggest adding data reliability test like Cronbach’s alpha.
g. there are several red flags on how the methodology has been explained.
6. The results in the analysis are discussed waguely and there is a much room for imrovemnet. In its current form it gives a feeling of simple statistical analysis of collected sample.
7. It is highly recommended to add research limitations and implications in conclusion as a separate heading.
8. In general, the use of the English language is fine. However, proofreading is required; especially, the sentence structure could have been better on several occasions.
I wish the author(s) good luck with the revision and resubmission.
Reviewer 2 Report
To investigate risk factors for collisions and near-miss incidents caused by drowsy bus drivers, the authors conducted a survey to collect the data and analyzed the data to make conclusions. However, there is no description of the study design. So I cannot make comments on their overall scientific soundness. Also, there is no detailed definition of the event group. If it is based on self-evaluated drowsiness, a sensitivity analysis is required. Another issue is that they collected more than one responses for some drivers. In the analysis, they just pooled the data together, which is not right as these samples are not independent.
Reviewer 3 Report
The manuscript presents an examination study of risk factors caused by drowsy bus drivers. The study was based on questioner survey of the 301 bus drivers of a bus company in Japan.
The manuscript is well written and well organized. The contribution could be a useful addition. In the opinion of this reviewer, it is recommended for publication considering the following comments.
- The significant concern is the date of study. The survey has been conducted in 2014. In the last years, there has been sharp development in the technology related to transportation and buses. In turn, the risk factors are expected to be affected accordingly and even new factors could appear. -The major debate become is how the conclusion of this study is affected. It is recommended to discuss, explain, and justify in more details.
- The introduction can be improved to include further related literature even if it was not focused on bus drivers.
- The responses were obtained from 519 of the 609 employees. However, the analyses were conducted only on the on drivers’ responses. Why did the other employee were questioned? Could the authors comment.
- There is confusing in the number of Near-miss incidents in the month preceding. Is the number of drivers 299 or 301? please clarify.
- Defining the importance of the risk factors is kind of decision-making process. The authors are recommended to adopt fuzzy analytical hierarchy process to define the importance weights of the risk factors. In this regard, it is worth referring to: Underground Space. 3(2018) 243-249.
Round 2
Reviewer 1 Report
Dear Authors,
Thank you for taking out time and reworking your paper. It looks in much better place than before.
Authors have moderately incorporated the suggestions, however, I still feel the need to expend the literature and add depth to it. I would suggest the following papers to be considered in adding value to your work.
- Azmat, M.; Kummer, S.; Moura, L.T.; Gennaro, F.D.; Moser, R. Future Outlook of Highway Operations with Implementation of Innovative Technologies Like AV, CV, IoT and Big Data. Logistics 2019, 3, 15. https://doi.org/10.3390/logistics3020015
- Wintersberger, S.; Azmat, M.; Kummer, S. Are We Ready to Ride Autonomous Vehicles? A Pilot Study on Austrian Consumers’ Perspective. Logistics 2019, 3, 20. https://doi.org/10.3390/logistics3040020
- Lam, A. Y., Leung, Y. W., & Chu, X. (2014, November). Autonomous vehicle public transportation system. In 2014 International Conference on Connected Vehicles and Expo (ICCVE) (pp. 571-576). IEEE.
Moreover, there is a need to address the procedure of data collection. Did you use snow bowling technique, sharpshooter strategy, or sho-gun strategy for this survey or briefly highlight your approach.
After addressing the aforementioned comments, your paper should be good to go.
All the best.
Reviewer 2 Report
There are big flaws in design and data analysis, which makes the conclusion invalid. I listed three major ones. First, the event is defined based on an accident caused by drowsy driving happened in the proceeding month. The time window is too long. Since the authors are interested in drowsy driving, it is more sensible to look at what happened a couple of days ago. Second, the event is self-identified and they are prone to recall biases, which requires validation from other sources or sensitive analysis for recall errors. Third, the non-event is defined as all others, who do not lead to "an accident caused by drowsy driving". It is not reasonable to combine accidents caused by other reasons and non-accidents together into the non-event group, which just adds more noise to the association study. Under the existence of competing risks, a more subtle analysis is required.